

# TaZFP 23, a new Cys2/His2-type zinc-finger protein, is a regulator of wheat (*Triticum aestivum* L.) growth and abiotic stresses

Shunxing Ye[1] and Yuzhou Tang[2]

[1] College of Bioscience and Biotechnology, Hunan Agricultural University, Changsha, China
[2] College of Landscape Architecture and Art Design, Hunan Agricultural University, Changsha, China

## ABSTRACT

Wheat (*Triticum aestivum* L.) is an important food crop and one of the most important grains in the world. With the global climate change, wheat production is increasingly affected by abiotic stress, among which drought, salinity, and other factors have become the main abiotic stress factors restricting the efficient production of wheat. The C2H2-type zinc finger proteins are a common class of transcription factors in plants that play crucial roles in regulating plant growth and development as well as responses to stresses. In this study, the wheat C2H2-type zinc finger protein transcription factor *TaZFP23* was cloned. Its full-length coding sequence was 720 bp encoding 239 amino acids. TaZFP23 is a typical C2H2-type zinc finger protein. It contains two C2H2 zinc finger domains and an EAR motif, without a transmembrane domain. Promoter *cis*-acting element analysis suggested that TaZFP23 might function in abiotic stress responses and plant hormone signal transduction. Subcellular localization and transcriptional activity assays indicated that TaZFP23 encoded a nuclear protein without self-activation activity. Overexpressing TaZFP23 in *Arabidopsis thaliana* showed that it negatively regulated seed germination and plant growth under NaCl, mannitol, and ABA treatments. Additionally, *TaZFP23* overexpression under NaCl and drought stress in *Arabidopsis* resulted in lower expression levels of several stress-related marker genes compared to those in wild-type plants. This research provides a foundation for further elucidating the functions of C2H2-type zinc finger protein genes and offers promising candidate genes for the development of stress-tolerant wheat cultivars.

# INTRODUCTION

Wheat is a vital staple food crop that is predominantly cultivated in arid and semi-arid regions across the globe. Nonetheless, its production frequently encounters suboptimal conditions in natural environments, including drought, high temperatures, low temperatures, and salinity, which can result in reduced grain yield or even plant mortality (*Rong et al., 2014*). Therefore, understanding the potential mechanisms of wheat response to abiotic stress is crucial for wheat production. During evolution, plants developed

Corresponding author
Shunxing Ye,
sevelyn@stu.hunau.edu.cn

mechanisms to mitigate damage due to harsh environments by regulating gene expression patterns. Regulation is primarily mediated by transcription factors that activate or suppress the expression of a series of genes (*Wang et al., 2019*; *Zhang et al., 2016a*; *Zhang et al., 2016b*; *Zhang et al., 2016c*). Transcription factors are proteins that contain DNA-binding domains or can bind to *cis*-elements on eukaryotic gene promoters, thereby regulating the transcription of target genes. Zinc finger proteins, as the largest family of transcription factors, are widely present in plants, animals, and fungi, with some being plant-specific (*Kim et al., 2009*). C2H2-type zinc finger proteins represent the most extensively studied class of zinc finger proteins in plants and fulfill various biological functions by binding to nucleic acids or proteins. These functions include transcriptional regulation, RNA metabolism, and chromatin regulation (*Wolfe, Nekludova & Pabo, 2000*). The first TFIIIA type zinc finger protein (ZPT2-1) in plants was identified from petunias during the study of the regulatory mechanism of petal specific expression of the enolpyruvate shikimate-3-phosphate synthase (*EPSPS*) gene, which is an enzyme in the shikimate pathway that leads to anthocyanin (*Takatsuji, 1998*).

The C2H2-type zinc finger protein motif typically follows the pattern Cys-X2-4-Cys-X3-Phe-X5-Leu-X2-His-X3-5-His (X: any amino acid), where two pairs of conserved His and Cys residues coordinate with $Zn^{2+}$ to form a finger-shaped tetrahedron with a $\beta$-$\beta$-$\alpha$ structure (*Sun et al., 2019a*; *Sun et al., 2019b*; *Kiełbowicz-Matuk, 2012*; *Pabo, Peisach & Grant, 2001*). Plant C2H2-type transcription factors generally feature one to four zinc finger domains, distinguished by two notable characteristics. First, the spacing between adjacent zinc finger domains is longer and more variable compared to that in animal proteins. For instance, in Petunia ZPT2-11, the two adjacent zinc finger domains are separated by 65 amino acids, whereas in most animals, zinc finger structures are typically separated by much shorter intervals of six to eight amino acids (*Takatsuji, 1998*). Secondly, the C-terminal alpha-helix of C2H2-type transcription factors contains a highly conserved sequence, QALGGH, which may be associated with specific biological activities in plants (*Takatsuji, 1998*; *Liu, Khan & Gan, 2022*). Additionally, C2H2-type transcription factors typically possess characteristic domains such as nuclear localization signals (NLS), DLN-box, L-box, and DNA-binding domains. These domains may participate in subcellular localization, transcriptional regulation, protein interactions, and other cellular processes (*Takatsuji, 1999*). C2H2-type zinc finger proteins are instrumental in a variety of biological functions, including the regulation of plant growth and development, as well as resistance to abiotic and biotic stresses (*Han et al., 2021*; *Han et al., 2020*; *Cabot et al., 2019*).

Many studies have reported that C2H2-type transcription factors regulate responses to abiotic stress (*Han et al., 2020*). IbZFP1, encoding a Cys 2/His 2 zinc finger protein gene from sweet potato, enhances salt and drought tolerance in transgenic *Arabidopsis* and it was shown to modulate the expression of genes related to salt stress to enhance salt tolerance (*Nguyen et al., 2016*). C2H2-type zinc finger proteins enhanced plant stress resistance by regulating the levels of osmolytes. AtSIZ1 and ZFP3 of *Arabidopsis* enhanced salt tolerance by promoting the synthesis and accumulation of free proline (*Zhang et al., 2016a*; *Zhang et al., 2016b*; *Zhang et al., 2016c*; *Han et al., 2019*). ZFP179 and ZFP252 enhanced salt tolerance in rice by increasing proline and soluble sugar levels and

regulating the expression of osmolyte synthesis-related genes such as *OsP5CS*, *OsLea3*, and *OsDREB2A* (*Sun et al., 2010*; *Xu et al., 2008*). OsDRZ1 enhanced resistance to drought stress by increasing proline levels and reducing ROS accumulation in rice (*Yuan et al., 2018*). C2H2-type zinc finger proteins were initially validated for their role in regulating plant stress responses by interacting with stress response factors and participating in the abscisic acid (ABA) signaling pathway. This interaction directly modulates the expression of downstream target genes, thereby contributing to the adaptive mechanisms of plants under stress conditions (*Han et al., 2019*). ABA and low temperature induced the expression of *SCOF-1*, which enhanced cold tolerance in transgenic *Arabidopsis* when overexpressed (*Kim et al., 2001*). By interacting with the bZIP transcription factor SGBF-1, SCOF-1 promoted the binding of SGBF-1 to ABRE *cis*-elements, increased *COR* expression, and enhanced plant cold tolerance (*Kim et al., 2001*). Overexpression of IbZFP1 enhanced drought and salt tolerance in *Arabidopsis* by increasing levels of ABA, proline, and soluble sugars, while reducing levels of malondialdehyde and $H_2O_2$ (*Wang et al., 2016*). Overexpression of the rice gene *OsMSR15* in *Arabidopsis* increased drought tolerance and ABA sensitivity, as well as enhancing the expression of stress-responsive genes *DREB1A*, *P5CS1*, *RD29A* under drought stress (*Zhang et al., 2016a*; *Zhang et al., 2016b*; *Zhang et al., 2016c*).

As of now, no reports have documented the involvement of the *TaZFP23* gene in regulating plant growth and abiotic stress. In this study, we utilized bioinformatics methods to predict the physicochemical properties, conserved domains, signal peptides, and *cis*-elements of TaZFP23. Additionally, we analyzed the subcellular localization of the TaZFP23 protein by transiently expressing *pCAMBIA2300-GFP::ZFP23* in tobacco leaves using *Agrobacterium*-mediated transformation.The SignalP online analysis of the TaZFP23 protein revealed the absence of a signal peptide, further indicating that it is a non-transmembrane protein. Subcellular localization predictions using Cell-PLoc 2.0 suggest that the protein encoded by TaZFP23 is localized in the nucleus. Additionally, the effects of NaCl and mannitol on seed germination were assessed in wild-type *Arabidopsis* as well as in lines overexpressing TaZFP23 (OE-5 and OE-31). The results demonstrated that both the seed germination and seedling emergence rates in the TaZFP23 overexpressing lines were significantly lower than those in the wild-type plants. These findings suggest that TaZFP23 negatively regulates seed germination under drought and salt stress conditions.

## MATERIALS AND METHODS

### Plant material and growth conditions

The experimental materials used in this study included wheat (*Triticum aestivum* L. Fielder), *Nicotiana benthamiana*, and *Arabidopsis thaliana* (ecotype Columbia 0, Col-0). Wheat and *Arabidopsis* were grown in a growth chamber at 22 °C with a photoperiod of 16 h light/8 h dark and a humidity of 80%. *Nicotiana benthamiana* was grown in a growth chamber at 25 °C with a photoperiod of 16 h light/8 h dark and a humidity of 80%.

After surface sterilization and cold stratification at 4 °C for 2–3 days, seeds of wild-type *Arabidopsis* (WT) and *Arabidopsis* overexpressing TaZFP23 were sown on 1/2 MS agar plates. Approximately 7–10 days later, seedlings were transferred to potting soil and placed
in a growth chamber at 22 °C. After 14 days, uniformly growing seedlings from each line were subjected to stress treatments. The stress treatments included watering with 350 mM NaCl solution every 4 days, drought stress (cessation of watering), and normal growth conditions. After 10 days of stress treatment, basal leaves from each line were harvested, rapidly frozen in liquid nitrogen, and stored at −80 °C for subsequent physiological, biochemical, and molecular analyses. Survival rates were recorded after 14 days of stress treatment. The experiment was conducted with three replicates, each consisting of 20 seedlings.

## RNA extraction and cDNA synthesis

Total RNA was extracted from wheat using the RNA isolater Total RNA Extraction Reagent from Vazyme Biotech. The RNA concentration was measured prior to storage at −80 °C using a Nanodrop 2000 instrument. For reverse transcription, the extracted RNA was treated with the HiScript II Reverse Transcriptase Kit from Vazyme Biotech, and cDNA synthesis was carried out following the kit's protocol.

## Obtaining and cloning of *TaZFP23* gene sequence in wheat

The coding region, promoter, and amino acid sequence of the wheat *TaZFP23* gene were downloaded from the Ensembl Plants database (http://plants.ensembl.org/). Specific cloning primers for *TaZFP23* were designed using Primer 5 software: *pCAMBIA2300-TaZFP38F*:ggtacccggggatcctctagaATGGCCGTGGAGGCGGTTCTTGAA; *pCAMBIA2300-TaZFP38R*:agctcctcctcctcctctagaCGCCGCGAGCATGAGCCTCG.

## Bioinformatics analysis of TaZFP23 protein in wheat

The amino acid sequence of TaZFP23 protein was submitted to ProtParam (http://web.expasy.org/protparam/) to predict its physicochemical properties. The conserved protein domains of TaZFP23 were analyzed using the SMART database (http://smart.embl-heidelberg.de/). The hydrophobicity of the TaZFP23 protein was predicted using ProtScale (https://www.expasy.org/resources/protscale). The transmembrane structure domains of TaZFP23 protein were predicted using the TMHMM-2.0 tool (https://services.healthtech.dtu.dk/services/TMHMM-2.0/). Subcellular localization of TaZFP23 protein was predicted using the Plant-mPLoc tool (http://www.csbio.sjtu.edu.cn/bioinf/plant-multi/). Signal peptide prediction for TaZFP23 was performed using SignalP 5.0 (https://services.healthtech.dtu.dk/service.php?SignalP-5.0). The 1,500 bp upstream sequence of the wheat TaZFP23 gene was submitted to PlantCARE (http://bioinformatics.psb.ugent.be/webtools/plantcare/html/) to predict *cis*-acting regulatory elements.

Subsequently, homologous protein sequences of TaZFP23 from various species were retrieved through alignment in Ensembl Plants. These sequences, along with the amino acid sequence of TaZFP23, were aligned using MUSCLE for sequence alignment. An evolutionary tree was constructed using MEGA7, employing the neighbor-joining method to analyze the evolutionary relationships among the TaZFP23 homologs from different species.

## Gene construct, transformation and expression analysis and transcription self activation

After purifying the PCR-amplified target fragment of *TaZFP23*, it was ligated into the XbaI-digested *pCAMBIA2300-GFP* vector for recombination. A one-step cloning kit from Novogene Corporation (Nanjing, China) was employed to construct the fusion expression vector. Subsequently, the plasmid was transformed into *Agrobacterium tumefaciens*.

Tobacco plants were cultivated in a growth chamber. After approximately two days, they were ready to be used for transient expression experiments. The bacterial culture was adjusted to an OD600 of approximately 0.3 by adding injection buffer. Using a syringe, the bacterial culture was injected into tobacco leaves and excess culture was carefully removed. The injected tobacco leaves were then incubated in darkness for two days. Subsequently, the injected tobacco leaf epidermis was examined to determine the subcellular localization of the targeted protein.

Yeast single colonies were inoculated into five mL of SD/-Trp liquid medium and incubated overnight at 30 °C with shaking at 180 rpm. 200 μL of the culture was transferred to a 1.5 mL microcentrifuge tube and centrifuged at room temperature at 12,000 rpm for 15 s. The supernatant was discarded, and 500 μL of sterile water was added to the pellet, followed by vortexing to resuspend the cells. This washing step was repeated twice. The OD600 of the yeast culture was adjusted to 1, and 3 μL of the culture was spotted onto SD/-Trp-Ade-His agar plates. The plates were then incubated at 30 °C for 4 days.

## The germination rate of seeds under NaCl mannitol and ABA treatments

Seeds of two homozygous lines, one overexpressing *TaZFP23* and the other a wild-type *Arabidopsis*, were harvested concurrently and subjected to surface sterilization and cold stratification at 4 °C for 2–3 days. The seeds were then sown on 1/2 MS solid medium containing varying concentrations of NaCl (0 mM, 80 mM, 120 mM), mannitol (0 mM, 200 mM, 250 mM), and ABA (0 μM, 0.5 μM, 1 μM). The plates were placed in a plant growth chamber at 22 °C under a 16-hour light/8-hour dark cycle for normal growth. Daily observations of the germination rate should be conducted using a microscope (SMZ-171; Motic). Each treatment group consists of three replicates, with 36 seeds per replicate. The eyepiece and objective lensof microscope should be adjusted as needed based on the specific circumstances.

## Measurements of survival rates and contents of proline and malondialdehyde

Before measuring the survival rates and contents of proline contents and malondialdehyde (MDA), *Arabidopsis* (two-week-old) seedlings were treated until clear differences in wilting were observed under drought conditions. About 0.1 g of plant leaves were taken, and MDA and proline contents were determined using an assay kit (Sangon Biotech, Shanghai, China) based on the manufacturer's protocols. All measurements were carried out with three biological replicates, and statistical analysis was carried out using one-way ANOVA.

### Enzyme activities assay

For enzyme activities assay, the enzyme extraction was performed from leaves of plants treated under drought conditions. Superoxide dismutase (SOD), catalase (CAT), and ascorbate peroxidase (APX) were measured according to previously described methods using an assay kit (Sangon Biotech, Shanghai, China). All extractions were obtained with three biological replicates.

### Data analysis

In this study, physiological parameters and seed germination rates were quantified and analyzed statistically. The physiological indicators, including (specific indicators such as survival rate (%), Pro, MDA, POD, CAT, SOD), were measured and expressed as means $\pm$ standard deviation (SD). Seed germination rates were calculated as the percentage of seeds that successfully germinated over the total number of seeds planted. All measurements were carried out with three biological replicates. Statistical analyses were conducted using one-way analysis of variance (ANOVA) to compare differences among treatment groups, with $t$-test applied for multiple comparisons. bars = 5 cm, Error bars indicate SD. $*P < 0.05$, $**P < 0.01$, $***P < 0.001$.

## RESULTS

### Cloning of *TaZFP23* into pCAMBIA2300 and transformation of its GFP-fusion

The Q-type C2H2 zinc finger protein gene sequences from rice and *Arabidopsis* were subjected to BLAST analysis on PlantGDB, leading to the identification of the EST sequence of C2H2 zinc finger protein gene *TaZFP23* in wheat. The *TaZFP23* EST sequence was then aligned using Ensemble Plants to identify the specific sequence information (100% nucleotide similarity) of *TaZFP23* (*TraesCS5B02G076400*). The expression vector for the TaZFP23-GFP fusion protein was constructed (Fig. 1A). The full-length coding sequence (CDS) of TaZFP23 is 720 bp and it was cloned with high-fidelity enzymes using wheat cDNA as a template (Fig. 1B). The TaZFP23 CDS was subsequently inserted into the *pCAMBIA2300* vector through one-step cloning. The clone was transformed into DH5$\alpha$ and positive clones were identified by colony PCR (Fig. 1C). All positive clones displayed the expected gene size. Following Sanger sequencing, the alignment of the target sequence with the sequencing results led to the generation of the expression vector *pCAMBIA2300-GFP::TaZFP23* for the TaZFP23-GFP fusion protein.

### Analysis of the physicochemical properties of the TaZFP23 protein

The amino acid sequence of the TaZFP23 protein was submitted to the ProtParam website (https://web.expasy.org/protparam/) for the prediction of its physicochemical properties (Table 1). It has a relative molecular weight of 24.43 kDa, a total number of 3,394 atoms, an isoelectric point of 7.07, a molecular formula of $C_{1045}H_{1686}N_{314}O_{337}S_{12}$, and an instability index of 57.38. These parameters suggested that the protein encoded by *TaZFP23* is an unstable one. The protein's lipid solubility coefficient is 68.12 with a total average hydropathy index of $-0.283$, indicating that the TaZFP23 protein is likely a hydrophilic protein.

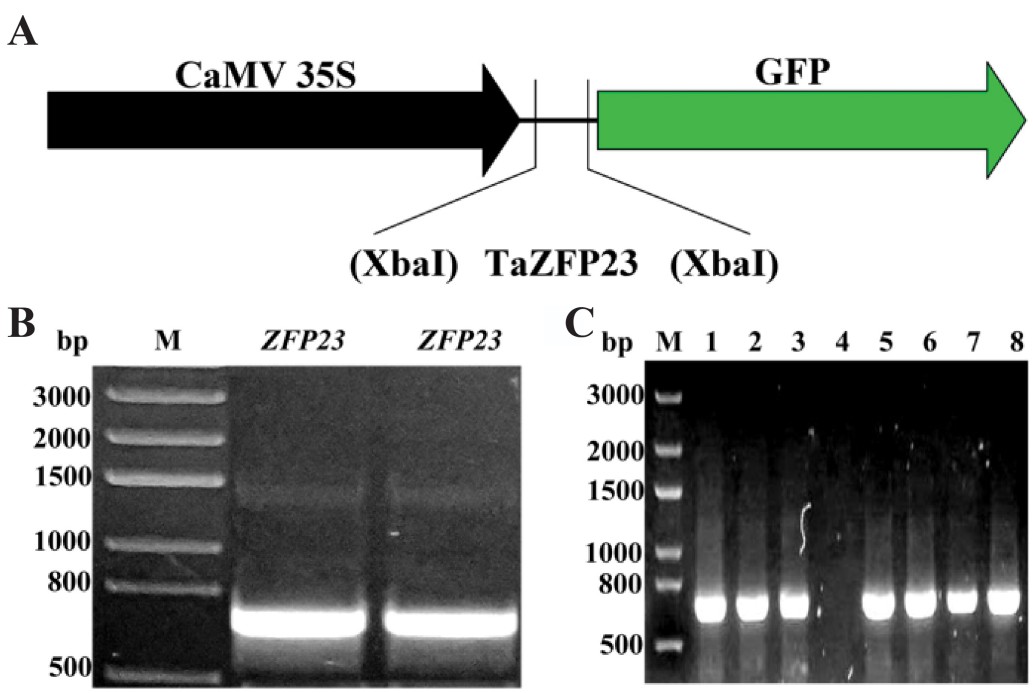

**Figure 1** **TaZFP23 construction and transformation of *pCAMBIA2300-GFP::TaZFP23* vector.** (A) TaZFP23 construction and transformation of *pCAMBIA2300-GFP::TaZFP23* vector; (B) TaZFP23 gene clone electrophoresis figure; (C) PCAMBIA2300 GFP:TaZFP23 colony PCR electrophoresis figure.

**Table 1 Physical and chemical properties of TaZFP23 protein.**

| Transcription factor | Molecular weight (KD) | Theoretical pI | Total number of atoms | Formula | Instability index | Aliphatic index | Grand average of hydropathicity (GRAVY) |
|---|---|---|---|---|---|---|---|
| TaZFP23 | 24.43 | 7.07 | 3,394 | $C_{1045}H_{1686}N_{314}O_{337}S_{12}$ | 57.38 | 68.12 | −0.283 |

## Analysis and prediction of TaZFP23 protein domain, hydrophobicity, transmembrane region, signal peptide, and subcellular localization

The structural domain analysis of the TaZFP23 protein was conducted using SMART (Simple Modular Architecture Research Tool, http://smart.embl-heidelberg.de). This analysis showed that the TaZFP23 protein contains two C2H2 zinc finger domains at positions 85–107 and 147–169, indicating that TaZFP23 belongs to the C2H2-type transcription factor (Fig. 2A).

Hydrophobicity plays a crucial role in studying the spatial and transmembrane structure of proteins. The hydrophobicity analysis of the TaZFP23 protein is shown in Fig. 2B, with most sequence positions scoring below 0. There is a significant peak at amino acid positions 40–50, suggesting that the protein is likely hydrophilic. Transmembrane domains of proteins are typically hydrophobic regions that play a significant role in maintaining their tertiary structure. TMHMM analysis of the TaZFP23 protein revealed that it does

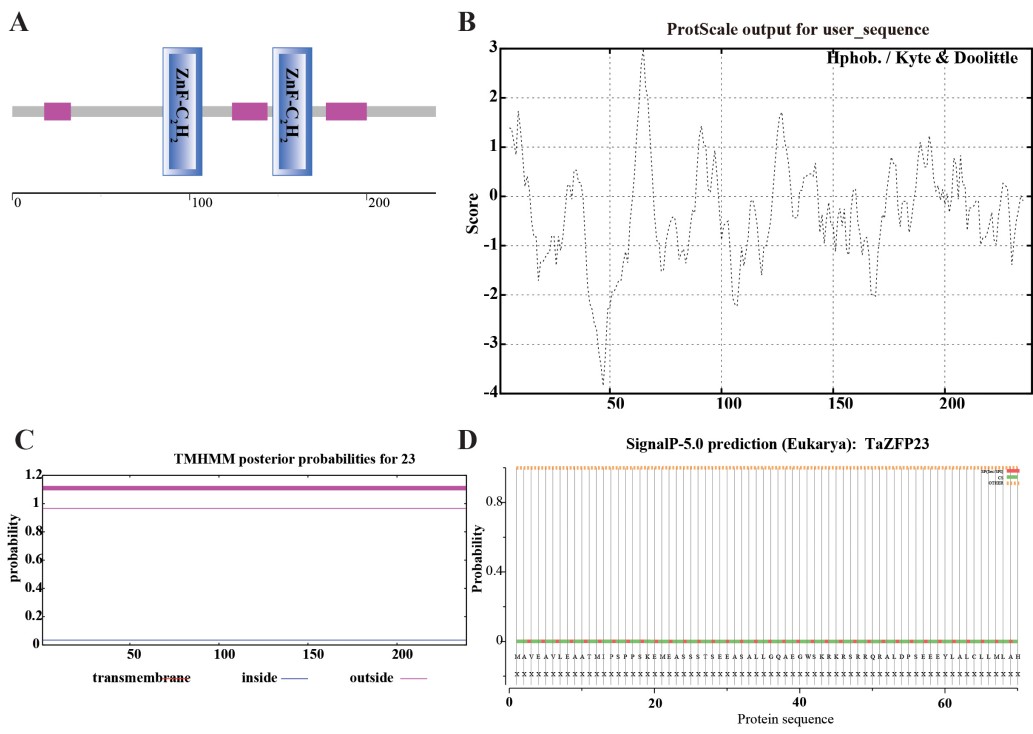

**Figure 2** **Analysis and prediction of TaZFP23 protein domain, hydrophobicity, transmembrane domains, signal peptide, and subcellular localization.** (A) TaZFP23 protein domain; (B) hydrophilicity; (C) transmembrane domains; (D) signaling peptides.

not have any transmembrane regions (Fig. 2C), indicating that it is a non-transmembrane protein.

## Promoter analysis of TaZFP23 gene

Transcription factors can regulate transcription by binding to *cis*-elements in the promoter sequences. The *cis*-elements in the upstream 1,500 bp promoter sequence of *TaZFP23* predicted using PlantCARE included ABRE, MBS, CGTCA-motif, LTR, TGA-element, G-box, ARE, and others (Table 2). These *cis*-elements are closely related to hormone and abiotic stress responses, suggesting that *TaZFP23* may play a role in abiotic stress responses and plant hormone signal transduction.

## Evolutionary analysis of TaZFP23 protein

Based on data in Ensemble Plant, homologous proteins of TaZFP23 were identified in rice, maize, barley, *Arabidopsis*, potato, tomato, and soybean. The results of the phylogenetic analysis reveal that the evolutionary relationship of TaZFP23 protein is closer to those in rice, barley, and maize, all of which belong to the Poaceae family (Fig. 3). The sequence similarity alignment showed that TaZFP23 and its homologous proteins all contain two C2H2 zinc finger domains and an EAR motif (Fig. 4). Furthermore, the C2H2 zinc finger domains of *TaZFP23* protein harbor a highly conserved QALGGH sequence, suggesting its classification into the Q-type C2H2 zinc finger protein subfamily.

**Table 2  cis-elements of *TaZFP23* gene promoter.**

| Element | Number | Function |
|---|---|---|
| ABRE | 5 | ABA response *cis*-element |
| MBS | 2 | Drought response *cis*-element |
| CGTCA -motif | 2 | MeJA response *cis*-element |
| G-box | 5 | Light response *cis*-element |
| TGA -element | 1 | Auxin response *cis*-element |
| LTR | 1 | Low temperature response *cis*-element |
| ARE | 2 | Anaerobic inducible response *cis*-element |

## Subcellular localization and transcriptional activity analysis of TaZFP23

Tobacco leaves injected with the expression vector carrying the fusion of TaZFP23 and GFP protein (*pCAMBIA2300-GFP::TaZFP23*) were used as the experimental group, while tobacco leaves injected with the control vector containing GFP tag (pCAMBIA2300) were used as the control group to detect the subcellular localization of the TaZFP23 protein. As shown in Fig. 4A, green fluorescence was observed throughout the cells in the control group, while green fluorescence was observed only in the nucleus of cells expressing TaZFP23, indicating that TaZFP23 protein is localized in the nucleus.

As a member of the Q-type C2H2 transcription factor family, TaZFP23 was examined for its potential transcriptional activation activity by cloning it into the pGBKT7 vector. Both the empty pGBKT7 vector and the recombinant plasmid pGBKT7-TaZFP23 were transformed into the yeast strain AH109 using the PEG/LiAc transformation method. Their activation activity was assessed on SD-III (SD/-Trp-His-Ade) deficient medium. As illustrated in Fig. 4B, yeast cells transformed with the empty pGBKT7 vector and pGBKT7-TaZFP23 were able to grow on SD-I (SD-Trp/Ade) medium. However, colonies were absent on the SD-III deficient medium, suggesting that the TaZFP23 protein may not possess self-activation activity.

## Germination rates of *TaZFP23* overexpressing *Arabidopsis* seeds under some abiotic stress treatments

Physiological indicators such as seed germination rate and seedling emergence rate are routinely used to assess stress tolerance ability of plants. The germination rates of wild-type *Arabidopsis* (WT) and *Arabidopsis* overexpressing *TaZFP23* (OE-5 and OE-31) were assessed under different concentrations of NaCl (80 mmol/L, 150 mmol/L), mannitol (200 mmol/L, 250 mmol/L), and ABA (0.5 μmol/L, 1 μmol/L) treatments. Results from the assessments showed that under normal conditions, there was little difference in the germination rates among WT, OE-5, and OE-31 lines. However, under NaCl treatment, significant differences in germination rates were observed between WT and OE-5, OE-31 lines, with the most significant differences seen on the 2nd day with 80 mmol/L NaCl treatment and on the 3rd and 4th days with 120 mmol/L NaCl treatment (Figs. 5A and 5B). These results suggested that overexpression of *TaZFP23* reduced the salt tolerance of *Arabidopsis*.

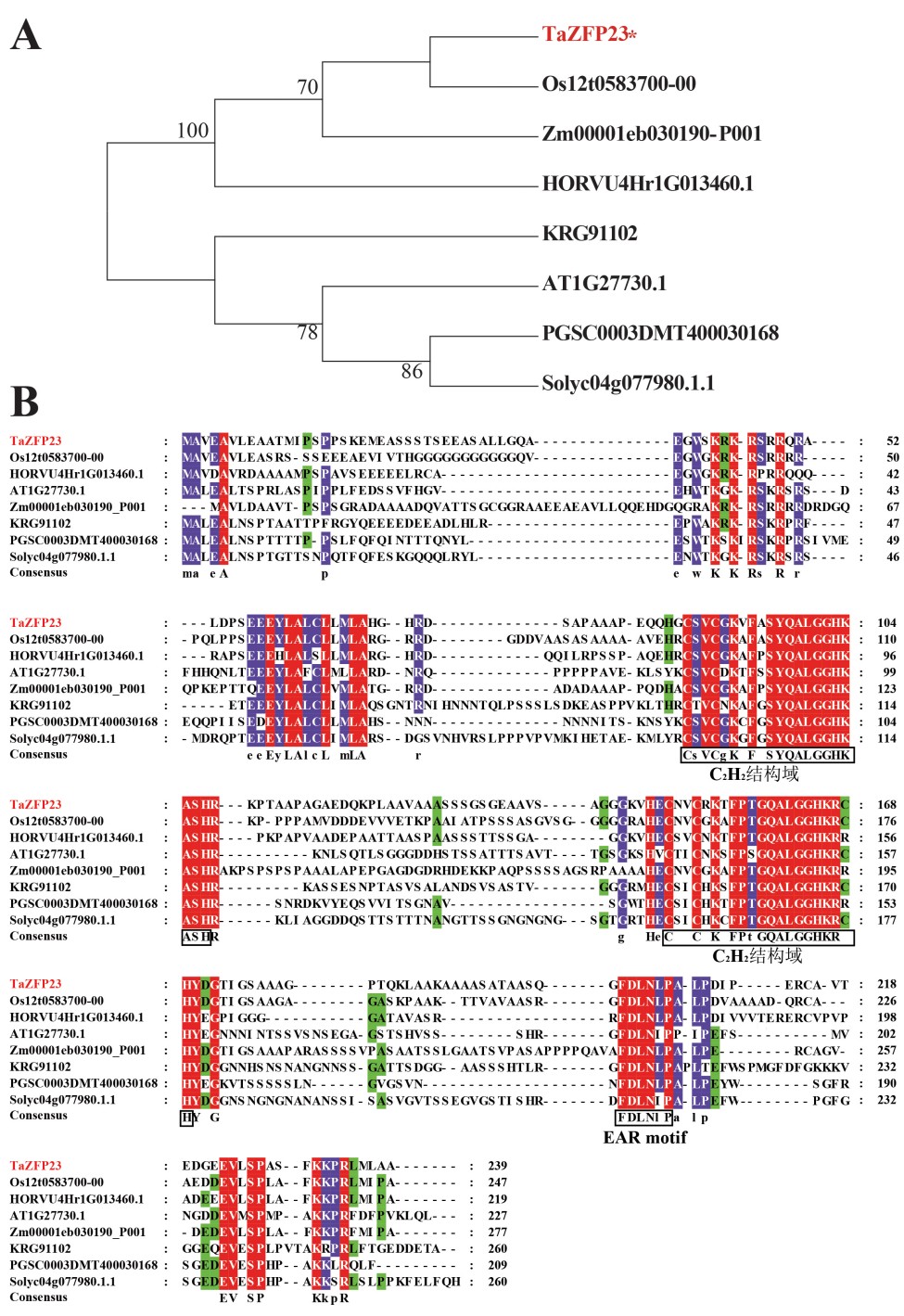

**Figure 3** **Evolution analysis and sequence alignment of TaZFP23 with homologous proteins from different species.** (A) Evolutionary analysis of TaZFP23 homologous protein system; (B) TaZFP23 homologous protein amino acid sequence alignment. Note: Wheat, TaZFP23; Rice, Os.

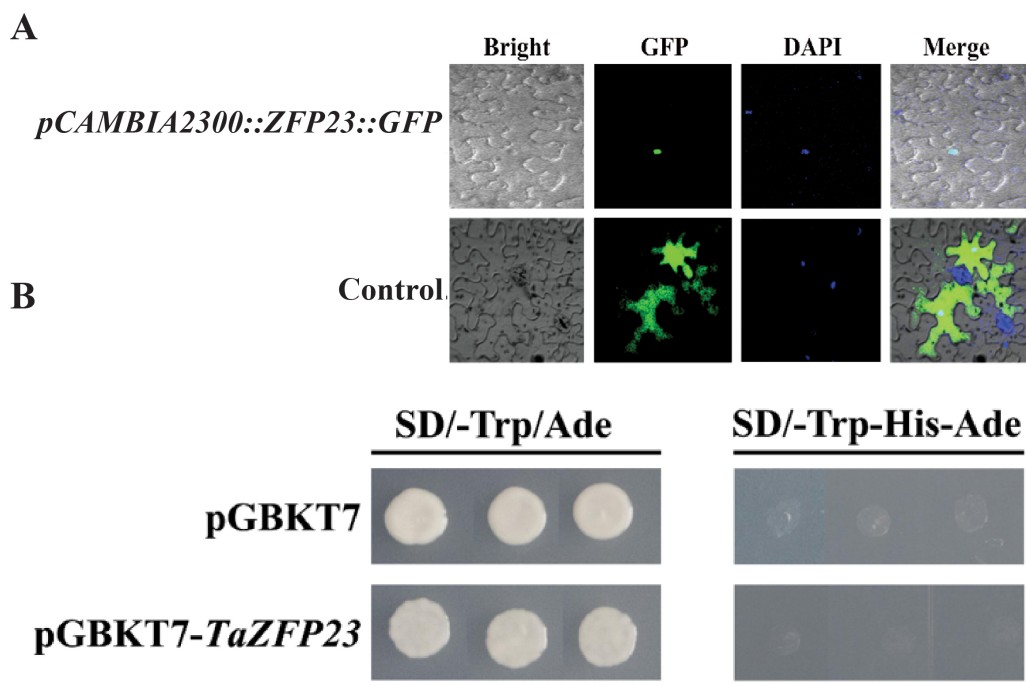

**Figure 4  Subcellular localization and transcriptional self activation activity analysis of TaZFP23 protein.** (A) Subcellular localization of TaZFP23 protein; (B) transcriptional self activation activity analysis of TaZFP23.

Under mannitol treatment, the germination rates of WT, OE-5, and OE-31 lines were all inhibited. The most significant differences were observed on the 2nd day with 200 mmol/L and 250 mmol/L mannitol treatments (Figs. 5C and 5D), indicating a negative regulatory role of TaZFP23 on seed germination under drought conditions in *Arabidopsis*.

Following ABA treatment, the germination rate of WT was significantly higher than that of the TaZFP23 overexpressing lines. The most significant differences were observed on the 2nd day with 0.5 μmol/L ABA treatment and on the 3rd day with 1 μmol/L ABA treatment (Figs. 5E and 5F). These results showed that overexpression of TaZFP23 increased the sensitivity of seed germination to ABA in *Arabidopsis*.

### Stress resistance analysis of *TaZFP23* overexpressing *Arabidopsis* under NaCl and drought stress treatments

Comparisons in the growth status and survival rates between the wild-type (WT) and *Arabidopsis* lines overexpressing *TaZFP23* under salt stress (300 mM NaCl) treatment were conducted. Varying degrees of chlorosis were observed in the leaves of *Arabidopsis* plants after 14 days of treatments with 300 mM NaCl (Figs. 6A and 6B). Significant differences in survival rates between WT (82.1%) and OE-5 (43.8%) or OE-31 (36.5%) were detected, indicating that overexpression of TaZFP23 reduced the salt tolerance of *Arabidopsis*. Moreover, most WT plants retained green leaves after 14 days of drought stress treatment, whereas the majority of OE-5 and OE-31 plants exhibited leaf wilting and eventual death (Figs. 6C and 6D). The survival rate of WT was 70.8%, while the survival rates of OE-5 and

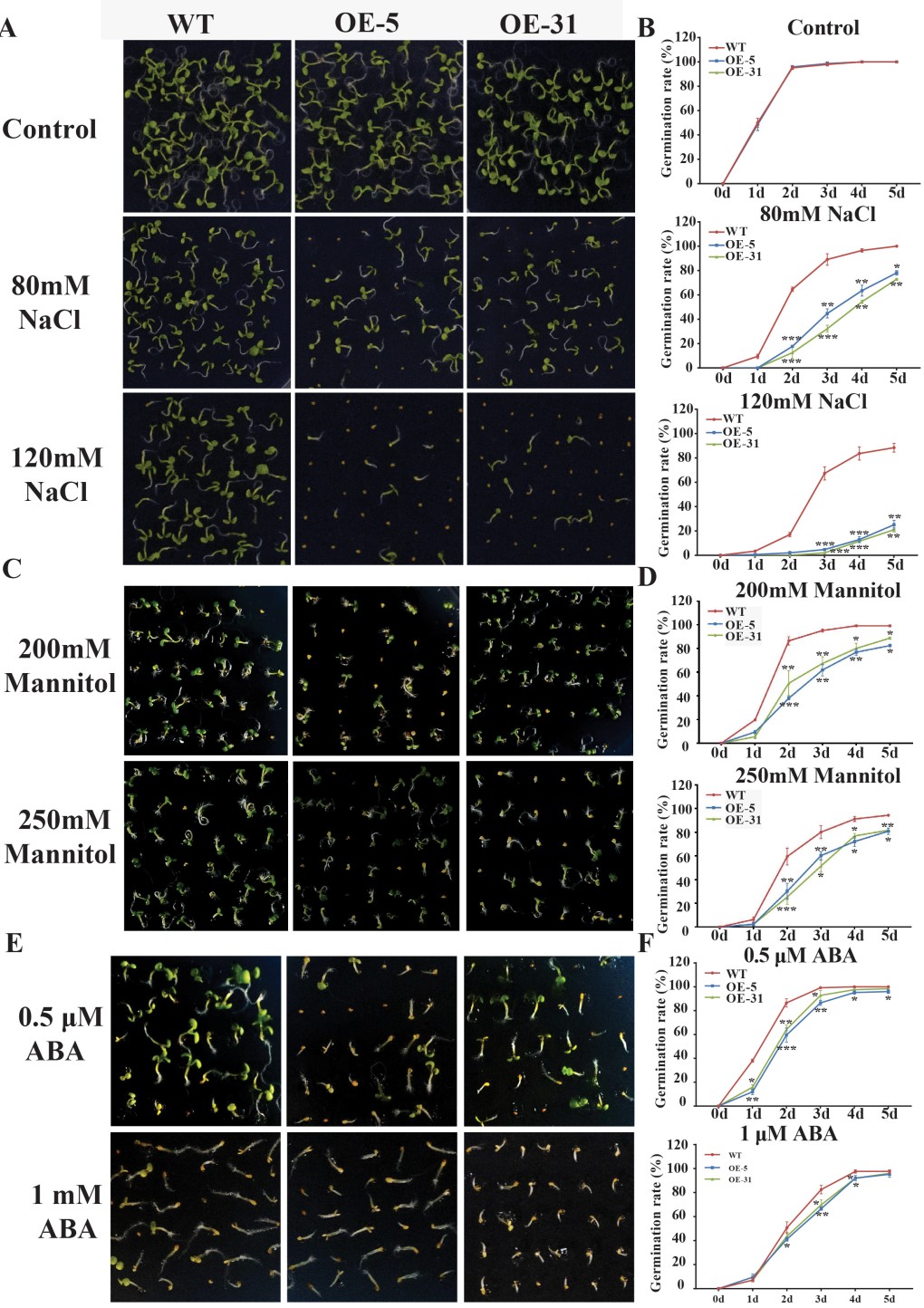

**Figure 5 Germination experiments of wild-type (WT) and TaZFP23 overexpressing *Arabidopsis* under abiotic stress treatment.** (A) The phenotype of overexpressing *Arabidopsis* seeds with WT and TaZFP23 under different concentrations of NaCl (80 mmol/L, 120 mmol/L) (continued on next page...)

**Figure 5 (...continued)**
treatment; (B) the germination rate under different concentrations of NaCl treatment; (C) the phenotype of overexpressing *Arabidopsis* seeds with WT and TaZFP23 under different concentrations of mannitol (200 mmol/L, 250 mmol/L) treatment; (D) germination rate under different concentrations of mannitol treatment; (E) the phenotype of overexpressing *Arabidopsis* seeds with WT and TaZFP23 under different concentrations of ABA (0.5 μ mol/L, 1 μ mol/L) treatment; (F) the germination rate under different concentrations of ABA treatment. Three repeated experiments, $*P < 0.05$, $**P < 0.01$, $***P < 0.001$.

OE-31 plants were 34.4% and 29%, respectively. The significant differences in survival rates suggested that overexpression of *TaZFP23* reduced the drought tolerance of *Arabidopsis*. It is well established that stress conditions can trigger lipid peroxidation in plants, leading to the accumulation of malondialdehyde (MDA) (*Fang et al., 2022*). Under non-stress conditions, no statistically significant differences in MDA content were detected among the wild-type (WT), OE-5, and OE-31 lines. However, when subjected to drought and salt stress, MDA levels in the leaves of the WT, OE-5, and OE-31 lines exhibited significant increases compared to the wild type (Fig. 6F). These results indicate that the overexpression of *TaZFP23* is correlated with elevated lipid peroxidation levels and more severe cellular damage in *Arabidopsis* under conditions of drought and salt stress.

Moreover, the activity of peroxidase (POD) and catalase (CAT) enzymes was analyzed. Theseare important protective enzymes in plant defense systems, reflecting metabolic changes in plants over time, and closely related to photosynthesis and respiration.

The activities of peroxidase (POD) and catalase (CAT) were investigated due to their established roles in photosynthesis and respiration, and because CAT activity serves as an indicator of metabolic levels and resistance to various stressors. The analysis revealed that enzyme activities were elevated in all three lines studied. However, the activities of both enzymes were significantly lower in the lines overexpressing *TaZFP23* compared to the wild type under drought and salt stress (Fig. 6H). Additionally, superoxide dismutase (SOD) activity increased in all three lines under these stress conditions. Notably, SOD activity in the leaves of the two lines overexpressing *TaZFP23* was significantly lower than that observed in the wild type (Fig. 6I). Collectively, these findings suggest that the overexpression of *TaZFP23* diminishes the tolerance of *Arabidopsis* to drought and NaCl stress.

## DISCUSSION

Drought, salinity, and temperature stress are primary environmental factors influencing the geographical distribution of plants, restricting crop yields, and jeopardizing food security. The rising frequency of extreme weather events will intensify the negative impacts of abiotic stress on agricultural production. How plants perceive and respond to environmental stress is a fundamental biological question. The abiotic stress factors typically lead to changes in plant morphology and metabolic levels, thereby impacting normal plant growth. Seed germination and survival rate are routinely used as growth indicators to measure their ability to tolerate drought and salinity. *Arabidopsis* lines overexpressing *ZAT6* exhibited higher germination rates under high salt conditions (*Liu et al., 2013*). The poplar gene

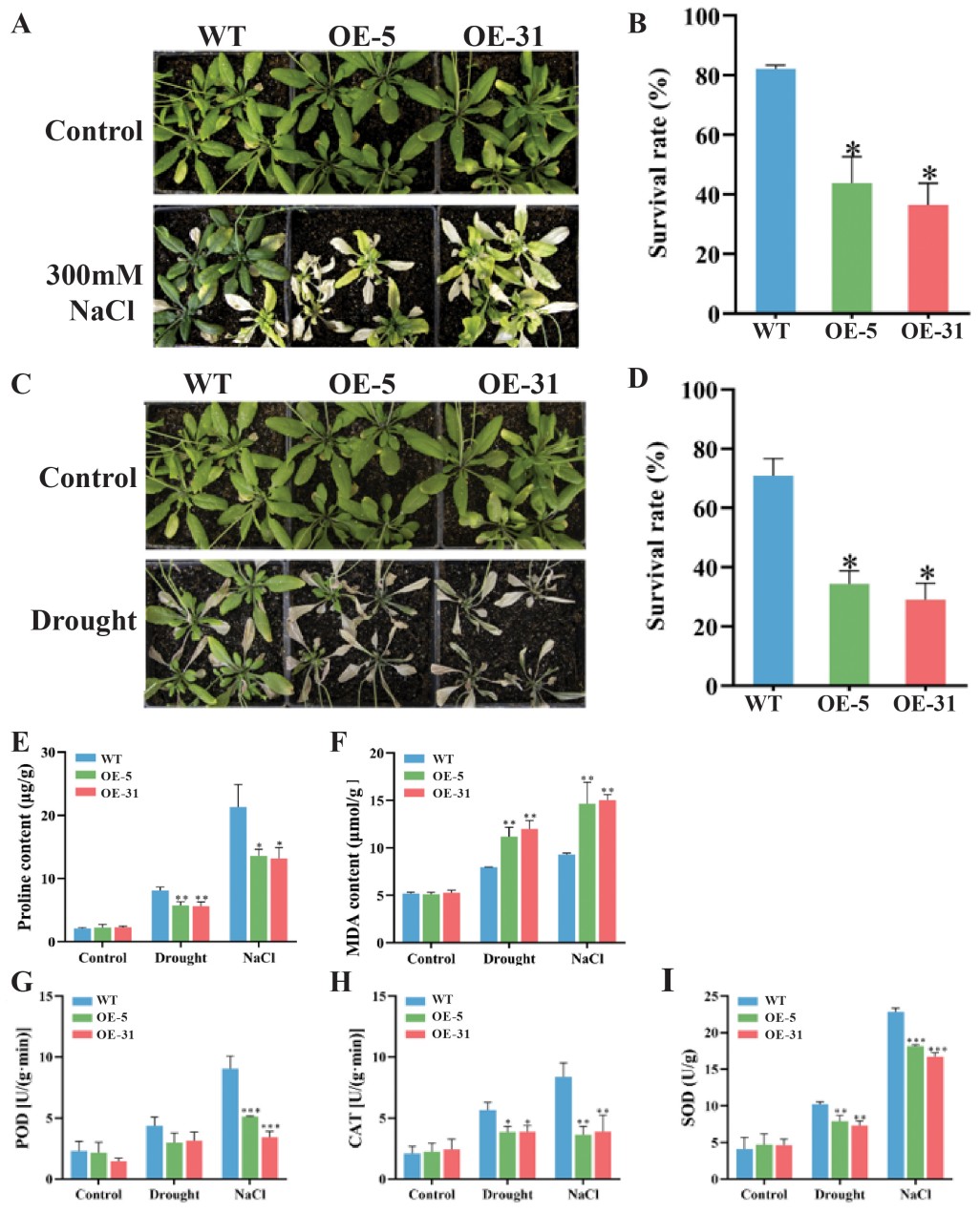

**Figure 6** Phenotypic experiments and physiological index analysis of *Arabidopsis* wild-type (WT) and TaZFP23 overexpressing *Arabidopsis* (OE-5, OE-31) under NaCl and drought stress treatments. (A) Phenotypes of WT, OE-5, and OE-31 under NaCl treatment; (B) survival rate statistics of WT, OE-5, and OE-31 under NaCl treatment; (C) survival rate statistics of WT, OE-5, and OE-31 under drought treatment; (D) drought of WT, OE-5, and OE-31 under NaCl treatment; (E) pro content; (F) MDA content; (G) the activity of peroxidase (POD); (H) the activity of catalase (CAT); (I) the activity of superoxide dismutase (SOD). Three repeated experiments, $*P < 0.05$, $**P < 0.01$, $***P < 0.001$.

*ZxZF* promoted root growth and increased survival rates under osmotic stress (*Zhang et al., 2016a*; *Zhang et al., 2016b*; *Zhang et al., 2016c*).

Members of ZFP family modulate abiotic stress responses in plants by regulating the transcription pattern of specific genes (*Sun et al., 2010*; *Chen et al., 2024*). Signal peptides can guide newly synthesized proteins through transmembrane transfer. Treatment of wild-type *Arabidopsis* and *TaZFP23* overexpressing *Arabidopsis* lines with ABA revealed that the germination rate and seedling emergence rate of WT were higher than those of the overexpressing lines, suggesting that *TaZFP23* might be involved in ABA signaling pathways. TaZFP23 may interact with abscisic acid (ABA)-related transcription factors to regulate the expression of genes associated with water stress. Specifically, although *Arabidopsis* is a widely used model organism in plant biology, its physiological characteristics and response mechanisms may differ from those of wheat. In order to comprehensively understand the biological functions of TaZFP23 in wheat and its response to abiotic stress, we will construct overexpression and deletion mutants of the *TaZFP23* gene, analyze the response of TaZFP23 to various abiotic stress conditions (such as salt, drought, *etc.*), and further elucidate the physiological functions of TaZFP23 in wheat. We will explore downstream signaling pathways and target genes related to TaZFP23 to elucidate its specific molecular mechanisms in plant stress response.

Proline is the most important osmoprotectant in plants, with stress-resistant plants accumulating more of it (*Ghosh et al., 2022*). Proline levels reflect plant ability to resist stress to some extent, with stress-resistant plants accumulating more proline following stress treatments. Significant differences in proline content in the leaves of WT, OE-5, and OE-31 lines of the control plants were not detected. However, under drought and salt stress conditions, proline contents in the leaves of WT, OE-5, and OE-31 lines increased to varying degrees. The proline contents in the leaves of OE-5 and OE-31 lines were significantly lower than that of the wild type. Additionally, drought and salt stress can disrupt the balance between the breakdown and production of ROS in plants, leading to the accumulation of excess ROS which damage cell membranes and biomolecules (*Nawaz & Wang, 2020*). The level of reactive oxygen species (ROS) in plants can be assessed by the accumulation of malondialdehyde. Environmental stresses disrupt the delicate balance of ROS, and superoxide dismutase (SOD) is crucial for mitigating excess ROS, thereby enhancing stress resistance. We evaluated SOD activity in various *Arabidopsis* lines under normal, drought, and salt stress conditions. Our results showed that the malondialdehyde content in *TaZFP23*-overexpressing *Arabidopsis* lines was higher than in the wild type under stress, suggesting that these lines may suffer more severe damage following stress treatments.

Enhancing the activity of antioxidant enzymes such as POD, CAT, and SOD can effectively remove excess reactive oxygen species (ROS) and mitigate stress-induced damage (*Bashri & Prasad, 2016*). AtRZFP has been shown to increase SOD and POD activities, reduce ROS accumulation, and positively regulate plant responses to salt and osmotic stress (*Zang et al., 2016*). Heterologous expression of chrysanthemum *CgZFP1* in *Arabidopsis* promoted the expression of ROS-scavenging-related genes like AtPOD and AtAPX1, thereby enhancing salt and drought tolerance (*Gao et al., 2012*). Increased

activities of POD, CAT, and SOD have also been associated with improved drought resistance in wheat (*Hameed, Goher & Iqbal, 2013*). Additionally, wheat TaZFP1 positively regulated responses to salt stress by enhancing CAT, SOD, and POD activities (*Sun et al., 2019a*; *Sun et al., 2019b*). In this study, SOD, CAT, and POD activities in *TaZFP23*-overexpressing *Arabidopsis* lines were significantly higher than those in the wild type under salt and drought stress. However, these findings suggest that TaZFP23 may inhibit protective antioxidant systems and impede ROS clearance, thereby reducing resistance to drought and salt stress. Overall, the results indicate that the TaZFP23 gene is involved in plant development and abiotic stress resistance in wheat.

To achieve a more comprehensive understanding of the biological functions of TaZFP23 in wheat and its responses to abiotic stress, we propose conducting functional complementation experiments. Investigating the synergistic effects of TaZFP23 in conjunction with other stress-related genes may elucidate its role under stress conditions. This approach could involve the suppression or overexpression of relevant genes to observe how these manipulations affect the function of TaZFP23 during stress. Additionally, employing systems biology methods to integrate diverse datasets, such as gene expression and metabolomics data, could facilitate the construction of a network model for TaZFP23 in stress responses. This would provide a more holistic understanding of its underlying mechanisms.

## CONCLUSIONS

In the study reported here, the wheat C2H2-type zinc finger protein transcription factor *TaZFP23* was cloned. It had a full-length coding sequence of 720 bp encoding 239 amino acids. TaZFP23 belongs to the typical C2H2-type zinc finger proteins, containing two C2H2 zinc finger domains and an EAR motif. It does not have a transmembrane domain. Promoter *cis*-acting element analysis suggests that TaZFP23 may function in abiotic stress responses and plant hormone signal transduction. Subcellular localization and transcriptional activity assays indicate that *TaZFP23* encodes a nuclear protein without self-activation activity. Functional analysis in *Arabidopsis thaliana* transgenic lines showed that *TaZFP23* negatively regulated seed germination and plant growth under NaCl, mannitol, and ABA treatments. Additionally, under NaCl and drought stress, *TaZFP23* overexpression in *Arabidopsis* resulted in lower expression levels of several stress-related marker genes compared to those in the wild-type plants.

### Funding

This work was supported by the National Natural Science Foundation of China (No. 82304652), the China Postdoctoral Science Foundation (No. 2022M721101), and the 2024 Hunan Province College Students' Innovation and Entrepreneurship Training Program (No. 2609). The funders had no role in study design, data collection and analysis, decision to publish, or preparation of the manuscript.

## Grant Disclosures

The following grant information was disclosed by the authors:
The National Natural Science Foundation of China: 82304652.
The China Postdoctoral Science Foundation: 2022M721101.
2024 Hunan Province College Students' Innovation and Entrepreneurship Training Program: 2609.

## Competing Interests

The authors declare there are no competing interests.

## Author Contributions

- Shunxing Ye conceived and designed the experiments, performed the experiments, analyzed the data, prepared figures and/or tables, authored or reviewed drafts of the article, and approved the final draft.
- Yuzhou Tang performed the experiments, authored or reviewed drafts of the article, and approved the final draft.

## Data Availability

The raw measurements are available in the Supplemental Files.

## Supplemental Information

Supplemental information for this article can be found online at http://dx.doi.org/10.7717/peerj.18956#supplemental-information.

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
