# Peer review of "TaZFP 23, a new Cys2/His2-type zinc-finger protein, is a regulator of wheat (Triticum aestivum L.) growth and abiotic stresses"

_PeerJ, doi:10.7717/peerj.18956_

## Round 0.1 · original submission · Major Revisions

Your manuscript needs major revisions as per the comments of the reviewers. Reviewers 1 and 3 have provided PDFs for you as well.

Reviewer 1 ·

Basic reporting

The basic reporting is clear and even though a few areas need refinement, professional English has been used throughout. The references are sufficient and solid. The figures are well presented. The results are relevant and well reported, although some modifications are needed.

Experimental design

The authors need to mention how many times experiments were repeated, further a paragraph on statistical tests used is missing.

Validity of the findings

no comment

Annotated reviews are not available for download in order to protect the identity of reviewers who chose to remain anonymous.

Reviewer 2 ·

Basic reporting

no comment

Experimental design

no comment

Validity of the findings

no comment

Additional comments

The manuscript “TaZFP23, a new Cys2/His2-type zinc-finger protein, is a regulator of wheat growth and abiotic stresses” by Shunxing Ye illustrates functions of C2H2-type zinc ûnger protein genes and provides
promising candidate genes for the development of stress-tolerant wheat cultivars based on their data. The manuscript obtained abundant data and the results are clearly presented.

Reviewer 3 ·

Basic reporting

First of all thanks for this valuable work.
I give all comments in the main manuscript file.
I think The English language should be improved ,

Experimental design

Please follow the comments in the main manuscript file
several things should be rewrite or revise and improve
Methods described must be rearranged and improve.

Validity of the findings

Conclusions are not well stated

Additional comments

Please discuss you findings with the published data specifically about ZFP

Annotated reviews are not available for download in order to protect the identity of reviewers who chose to remain anonymous.

·

Basic reporting

The basic reporting of the study provides a clear description of the experimental objectives, methods, and findings.

Experimental design

The overexpression of TaZFP23 in Arabidopsis thaliana provides a useful model to assess its functional role under stress conditions. However, the experimental design could be strengthened by developing transgenic lines of wheat (or related species) that overexpress TaZFP23. This would help ensure that the observed effects are not species-specific or limited to a single transgenic line, and would improve the relevance and reproducibility of the findings in the context of wheat or closely related crops. Incorporating such a strategy would significantly enhance the robustness and applicability of this work.

Validity of the findings

Gene Validation: Although the promoter analysis suggests a regulatory role, experimental validation through reporter assays or mutation of the predicted cis-elements would provide stronger evidence of TaZFP23's function in stress responses.

The report states that TaZFP23 does not show self-activation activity but does not specify the methods used to assess this. Was this through a yeast one-hybrid system or a different assay? Including more information on the assay conditions and controls would clarify this point.

Additional comments

1. The report does not mention any limitations of the study or areas for further investigation. Acknowledging potential limitations (e.g., the use of Arabidopsis as a model system for wheat) and suggesting future research directions would strengthen the overall reporting and provide a roadmap for further exploration of this transcription factor.

2. While the study provides useful information on the role of TaZFP23 in stress tolerance, it could benefit from a deeper discussion of the molecular mechanisms by which TaZFP23 exerts its effects. For example, how might TaZFP23 interact with other transcription factors or signaling pathways involved in stress responses?

3. Italicise all scientific names, genes names for eg, line No. 102 (Nicotiana benthamiana), line No. 107, 196, 200 etc. Correct throughout the manuscript.

4. Consistency in Terminology: There are occasional inconsistencies in terminology (e.g., “zinc finger domains” vs. “zinc finger motifs” or “cis-acting elements” vs. “promoter

---

## Round 0.2 · Minor Revisions

Your manuscript needs some remaining minor corrections

Reviewer 1 ·

Basic reporting

Their are grammatical errors throughout the manuscript. These could be corrected during the typesetting stage. But it has been indicated in the manuscript.

Experimental design

no comments

Validity of the findings

no comments

Additional comments

The discussion section still needs improvement as indicated in the comments.

Annotated reviews are not available for download in order to protect the identity of reviewers who chose to remain anonymous.

Reviewer 3 ·

Basic reporting

no comment

Experimental design

no comment

Validity of the findings

no comment

Additional comments

no comment

Annotated reviews are not available for download in order to protect the identity of reviewers who chose to remain anonymous.

·

Basic reporting

All comments have been thoroughly addressed in the revised manuscript. The manuscript is now ready for acceptance

Experimental design

no comment

Validity of the findings

no comment

Additional comments

no comment

---

## Round 0.3 · accepted · Accept

Your manuscript has been accepted after your final revisions.